# Credit Risk Prediction Based on Psychometric Data

**Eren Duman [1], Mehmet S. Aktas [1,\*] and Ezgi Yahsi [2]**

1   Computer Engineering Department, Yildiz Technical University, Istanbul 34320, Turkey;
    eren.duman1@std.yildiz.edu.tr
2   Research and Development Center, Aktifbank, Istanbul 34394, Turkey; ezgi.yahsi@aktifbank.com.tr
\*   Correspondence: aktas@yildiz.edu.tr

**Abstract:** In today's financial landscape, traditional banking institutions rely extensively on customers' historical financial data to evaluate their eligibility for loan approvals. While these decision support systems offer predictive accuracy for established customers, they overlook a crucial demographic: individuals without a financial history. To address this gap, our study presents a methodology for a decision support system that is intended to assist in determining credit risk. Rather than solely focusing on past financial records, our methodology assesses customer credibility by generating credit risk scores derived from psychometric test results. Utilizing machine learning algorithms, we model customer credibility through multidimensional metrics such as character traits and attitudes toward money management. Preliminary results from our prototype testing indicate that this innovative approach holds promise for accurate risk assessment.

**Keywords:** psychometric test; credit risk assessment; artificial intelligence; decision support systems; creditworthiness

## 1. Introduction

There is a need for decision support software that can predict the likelihood of a loan being repaid based on a customer's behavioral data. Credit risk management is a crucial concern for any organization, regardless of the circumstances, but it holds particular significance within the banking sector. This reality becomes even more apparent during periods of financial turmoil, as financial institutions face the potential of significant losses stemming from unpaid debts [1]. Not appropriately handling credit risks can also lead to more than just direct financial losses. This includes missed opportunities, costs associated with transactions, and expenses related to non-performing assets, in addition to the accounting losses. It has the potential to impact the bank's portfolio, which could result in liquidity risk, and in severe cases, it may have detrimental effects on both the financial sector and the economy as a whole [2].

Furthermore, the importance of measuring credit risk has significantly increased in today's changing and evolving lending landscape. Alongside the traditional practice of obtaining large loans, there is now a trend of providing smaller installment-based loans in retail stores, resulting in a significant escalation in the frequency of credit borrowing. This shift in lending practices has necessitated that financial institutions develop comprehensive methodologies for evaluating credit risk [3].

In this study, we develop a comprehensive methodology to address several pressing requirements for decision support software in the credit risk domain: The first requirement is for a system capable of evaluating credit risk and default probabilities for customers who lack adequate financial data. Our approach supplements traditional metrics with behavioral analytics, offering a more holistic understanding of an individual's creditworthiness. The second requirement targets financial inclusion, particularly in low-income economies where a vast segment of the population—numbering in the hundreds of millions—faces barriers to credit and banking due to insufficient financial history. In these cases, traditional

financial data are ineffective for assessing repayment capacity, leading to heightened risks and associated costs when lending without proper evaluation [4]. Our methodology aims to mitigate such risks through software-assisted evaluations. The third requirement focuses on improving decision-making accuracy when approving banking products. A common cause of financial losses is erroneous decisions made due to incomplete knowledge of a customer's behavior and attitudes. Our system incorporates psychometric testing to gain a more nuanced understanding of applicants' personal and cognitive attributes. This enables financial institutions to either charge risk-aligned interest rates or outright deny high-risk loan applications, thereby optimizing revenue potential.

The objective of this study is to determine the effectiveness of using psychometric tests in the credit lending process and to assist credit lending institutions in more accurately measuring their customers' creditworthiness. Within the scope of this study, a decision support system was developed to predict the credit risk of the customer. The following questions are listed under the research problem examined in this study: (1) how much do psychometric tests contribute to credit lending decisions compared to traditional credit assessment methods? (2) how can we analyze the behaviors that indicate a customer's creditworthiness? (3) how can a customer's credit risk be predicted based on feature vectors, and what machine learning algorithms can be used for this purpose?

Banks require a decision support system that can calculate the credit risk of customers, particularly those with no or incomplete financial history. To address this need, we outline the contributions of this study as follows: This study presents a novel approach by developing a software architecture that leverages customers' psychometric data by modeling them with machine learning techniques such as random forest classifier, decision tree, and logistic regression to enhance the accuracy of credit risk assessment. By analyzing factors such as responsibility, normalization of borrowing, self-discipline, self-sufficiency, emotional stability, impulsivity, and other psychometric attributes, this study seeks to expand upon the existing knowledge and explore these attributes' specific impact on credit risk assessment.

The organization of this manuscript is as follows: Section 2 overviews the fundamental concepts and the literature review. Section 3 outlines the proposed methodology. Section 4 elaborates on the prototype implementation and experimental research results. Section 5 discusses the conclusions of this study and proposals for future work.

## 2. Fundamental Concepts and Literature Review

### 2.1. Fundamental Concepts

Creditworthiness: Creditworthiness is a concept used to determine whether a borrower's loan request should be approved or not. This concept helps lenders assess the borrower's ability to repay the loan.

In measuring creditworthiness, various psychometric features that the borrower has, such as responsibility, importance given to material values, self-efficacy, self-control, and other relevant features, are taken into consideration. Self-efficacy evaluates an individual's confidence in their ability to complete a task. Research on self-efficacy indicates that individuals with elevated efficacy ratings possess ambitious aspirations, thus exhibiting greater perseverance and a stronger inclination to achieve their objectives [5]. Self-control plays a crucial role in the realm of consumer credit, especially when it comes to preventing consumer over-indebtedness. Beyond its broader implications, self-control encompasses the power to break free from detrimental patterns, manage anger, and conquer impulsive emotional urges. Within the context of financial transactions, self-control serves as a psychological foundation for recognizing the likelihood of excessive credit card consumption. By exercising self-control, individuals can navigate the potential pitfalls of overusing credit cards and make more informed decisions about their financial well-being [6]. In traditional creditworthiness measurement methods, characteristics consist of numerical features such as credit score, income, credit history, and employment history rather than psychometric qualities. However, there are customers who do not have a credit history or sufficient data

in this regard. Making a measurement with psychometric data will play a role in both calculating the credit risk of these customers and increasing the accuracy of traditional methods.

Supervised Machine Learning Techniques: Machine learning is a type of artificial intelligence that enables computer systems to learn and improve automatically without being explicitly programmed. Machine learning is a branch of artificial intelligence and computer science that focuses on the use of data and algorithms to imitate the way that humans learn, gradually improving its accuracy [7]. It involves using algorithms to identify patterns in data, learn from them, and make predictions or decisions based on those patterns. Machine learning encompasses three primary types: supervised learning, unsupervised learning, and reinforcement learning. Supervised machine learning involves generating a function that establishes a relationship between inputs and desired outputs. One common variant of supervised machine learning is the classification problem, where the learner aims to approximate the behavior of a function that assigns a vector to various classes based on observed input–output examples. In this study, we focus on utilizing supervised machine learning to predict creditworthiness. We utilize supervised machine learning algorithms like logistic regression, random forest, and decision tree [8].

Feature Vectors: Feature vectors play a crucial role in machine learning by providing a mathematical representation of object characteristics, both numeric and symbolic, in a format suitable for analysis. These vectors are fundamental across various domains within machine learning and pattern recognition. To enable processing and statistical analysis, machine learning algorithms often rely on numerical representations of objects. Feature vectors serve as the counterpart to explanatory variable vectors employed in statistical techniques such as linear regression. By utilizing feature vectors, machine learning algorithms can effectively extract meaningful patterns and insights from data.

The process of creating a feature vector involves selecting a set of relevant features or variables that can be used to describe the data point. These features can be either numeric or categorical and may be derived from a variety of sources, such as sensor readings, textual data, or image pixels. In this research, these features consist of psychometric features that impact a person's credit score. Once the feature vector has been created, it can be used as input for a machine learning algorithm or other data analysis technique. The goal of these methods is typically to learn a model or function that maps the feature vector to some output, such as a classification label or a numerical value.

### 2.2. Literature Review

Effective credit risk management is paramount in modern financial systems, especially in the banking sector. In this section, we explore various themes within the existing literature, ranging from traditional credit scoring models to emerging technologies and methodologies like machine learning and psychometrics.

Credit scoring has been a central topic for researchers aiming to improve financial decision-making processes. Traditionally, credit scoring models focus on numerical factors such as income, age, employment status, and existing debts [9,10]. However, studies by [11,12] indicate the limitations of these traditional models, particularly for applicants without a substantial credit history. A significant body of research indicates the limitations of relying solely on traditional metrics. The authors of [12,13] found that these methods often fail to consider the applicant's full financial capabilities and could be exclusionary, especially for low-income or young applicants. A growing trend in the literature is the integration of behavioral economics into credit scoring models. The authors of [14,15] demonstrated that psychological attributes like impulsivity, risk aversion, and financial literacy substantially impact creditworthiness. Machine learning technologies are increasingly being explored for their potential to revolutionize credit risk assessment. The authors of [16] showed that random forest algorithms could predict with high accuracy. Meanwhile, The authors of [17] applied decision trees to improve the robustness of credit scoring, albeit still using traditional financial indicators. As artificial intelligence is integrated into financial systems, ethical considerations emerge. Research by [18] underlines the importance of

ensuring these algorithms do not inadvertently discriminate against particular social or economic groups. Financial inclusion is a growing concern in both academia and policy, especially for low-income or emerging economies. The authors of [13] emphasized the need for alternative credit scoring mechanisms that could evaluate the creditworthiness of individuals with insufficient financial histories. The utilization of psychometric testing in assessing credit risk is a burgeoning field of study. The authors of [19] demonstrated the effectiveness of using psychometric factors, including responsibility and emotional stability, in predicting credit risk.

While advancements have been made in these individual areas, there is a notable gap in the literature concerning the integration of these various components into a unified, sophisticated credit risk assessment model. This research seeks to fill that gap by exploring the effectiveness of a holistic approach that combines traditional financial metrics, behavioral analytics, and machine learning algorithms.

The design and development of decision support system software based on the results of psychometric tests have been studied over the years. For example, in [20], the effects of customer personality traits on the predictability of creditability were evaluated by statistical analysis. Then, a creditability model was created using artificial learning algorithms. Various variables, such as customer personality traits, loan application information, and economic data, were used in the model. The methodology used in the article includes psychometric scales, statistical analyses, and machine learning algorithms.

Within the scope of [21], the personality traits of university students were measured using the Big Five Personality Inventory, and some data used in the loan application process were collected. The results showed that personality traits can be effective in predicting credibility. The methodology of the study includes correlation analysis, regression analysis, and factor analysis.

Within the scope of [22], the usability of the personality traits of loan applicants in the measurement of credibility is investigated. The researchers measured the personality traits of loan applicants using the Big Five Personality Inventory and also considered the criteria used to measure creditworthiness. As a result of the study, it was seen that personality traits can be effective in predicting credibility. The methodology of the study includes correlation analysis, factor analysis, and regression analysis.

Within the scope of [23], the contribution of psychological variables and scales suggested by economic psychology in predicting individuals' default is investigated. In the study, 555 participants were surveyed using a self-administered questionnaire that included various psychological variables and scales. These studies adopted the methodology of logistic regression, and they found some psychological and behavioral characteristics associated with the group of individuals in default, such as self-efficacy, being compulsive, and self-control.

New studies in financial behavior have looked at different psychological factors that affect paying back debt. These studies show the importance of looking at more than just a person's financial past [24]. Recent studies in credit risk assessment have emphasized the diverse factors contributing to individual credit risk. Balina and Idasz-Balina [25] examined the drivers of individual credit risk within the cooperative banking sector, highlighting the importance of localized financial conditions. Mhlanga [26] discusses the role of machine learning and artificial intelligence in enhancing financial inclusion in emerging economies through more accurate credit risk assessment. Doko et al. [27] proposed a credit risk model utilizing central bank credit registry data, reflecting the potential of institutional data in risk modeling. Meanwhile, Ganbat et al. [6] explored the impact of psychological factors on credit risk, particularly within the context of microlending services. Kanapickienė et al. [28] investigated macroeconomic determinants of credit risk, offering insights into the influence of wider economic factors on consumer credit. Much research has been done on how financial literacy affects buyer behavior [29]. On the other hand, Kahneman and Tversky's work on how people make decisions when they are faced with risk [30] is similar to the use of psychometrics to help reduce poverty [31]. Studies have also shown that age and gender

affect how much debt people have [32,33] and how they feel about new financial trends like cryptocurrency [34]. Personality traits have been linked to how much people save and borrow [35], and a multivariate study has shed light on the problem of having too much debt [36]. People know that behavioral biases and intuitive biases play a big role in financial choices [37] and [38]. Kahneman gives an overview of how decisions are made [39]. Much research has been done on the link between demographics, attitudes, and credit card debt in different cultures [40]. Also, it has been shown that financial understanding is a key factor in both financial planning and consumer behavior [41].

This study focuses on designing and developing a process for credit risk prediction utilizing behavioral data analysis. There exist studies focusing on building applications based on distributed computing and web services [42–44]. Different from this previous work, this study focuses on designing and implementing a process that can predict the credit risk situation by analyzing the users behavioral data. To show the usefulness of the proposed process, a prototype is implemented. There exist studies that focus on analyzing the quality of the software [45,46]. However, in this study, we leave out the software quality analysis for future work.There are studies that focus on understanding users' actions by analyzing the user-system interaction data [47–49]. This study analyzes the users' behaviors based on their psychometric test data.

## 3. Methodology

In this study, we introduce a business process for software, which is a decision support system designed for comprehensive credit risk management in the banking sector. Utilizing machine learning algorithms and psychometric testing, it assesses behavioral attributes to provide a more accurate and inclusive evaluation of an individual's creditworthiness. Figure 1 introduces the conceptual framework for a business process aimed at predicting credit risk through behavioral data analysis. This diagrammatic representation begins by capturing the initial data collection phase, where various behavioral and psychometric parameters are gathered from potential borrowers. Subsequent stages involve the processing of this data through advanced analytics, leveraging algorithms that assess and weigh the collected information, ultimately culminating in a credit risk score. With this framework, it is possible to obtain a more comprehensive picture of a borrower's profile that goes beyond typical financial constraints. This framework highlights a way of incorporating psychological measurements into traditional financial risk assessment. Below, we discuss the details of each module of this business process.

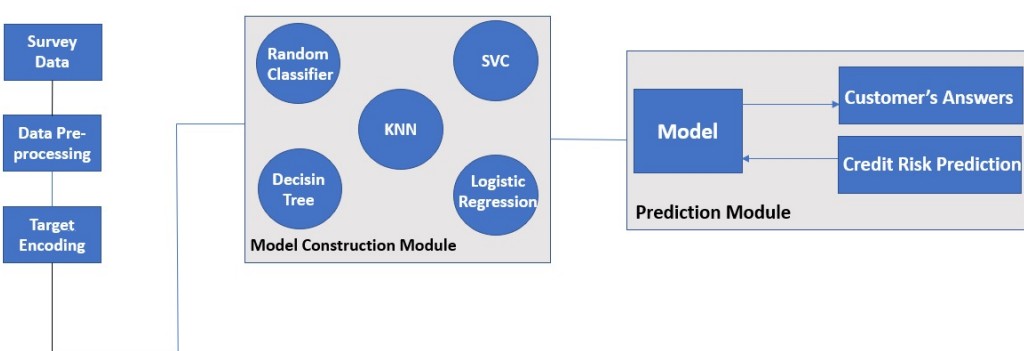

**Figure 1.** The proposed business process for credit risk prediction based on behavioral data analysis.

### 3.1. Data Pre-Processing Module

The dataset used in this study consists of an 18-item questionnaire. The raw data are in the form of a text file containing questions and their corresponding answers. The dataset includes 1 dependent feature and 14 independent features. The list of dependent and independent features and their descriptions is presented here. Dependent feature: credit risk—a measure of the likelihood that a borrower will not fulfill their commitment to repay

a loan or debt to a lender. Independent features: problem-solving ability—this attribute indicates how well a person is able to solve financial problems when faced with them; normalization of borrowing—this attribute reflects an individual's credit habits, where the habit or frequency of borrowing is a factor that should be taken into account when assessing an individual's credit risk; importance given to material values—the attribute refers to the values that shape a person's financial decisions, where the importance a person attaches to material values can affect credit risk because these values can influence debt repayment or habits; financially prepared—this attribute reflects how financially prepared a person is, where financial readiness is an important factor in determining a person's resilience to unexpected financial difficulties; importance given to comfort—this attribute reflects a person's comfort and lifestyle preferences that determine their financial decisions, where the importance given to comfort can affect one's credit risk because it can influence spending habits; self-discipline—reflects how an individual handles their financial responsibilities, where a person's self-discipline is a critical factor in terms of loan repayments and debt management; self-sufficiency—this attribute measures an individual's ability to meet their own needs, where a high value may indicate that the individual has financial independence and self-confidence; self-control: this attribute represents an individual's ability to control their financial spending and debt, where a low self-control score may negatively affect the management of debts and increase credit risk; impulsivity—this attribute measures an individual's impulsive financial decisions and risky spending, where a person with a high level of impulsivity is likely to make financial decisions without thinking and increase credit risk; emotional stability—this attribute reflects how an individual copes with financial stress and emotional reactions, where emotional instability can negatively affect one's ability to cope with financial problems; responsibility—this attribute measures an individual's ability to fulfill financial responsibilities, where a high level of responsibility can reduce credit risk; unconcerned—this attribute expresses how unconcerned the individual is towards problems, where a high level of apathy may indicate a lack of awareness of financial problems and increase credit risk; self-efficacy—this attribute reflects an individual's ability to solve financial problems and achieve financial goals, where a high level of self-efficacy can positively influence financial success; desire to impress people—this attribute measures an individual's desire to impress others or to show status, where a high level of desire to impress people can lead to unnecessary expenditure and financial risks, thus increasing credit risk.

We transformed the test responses into a structured tabular format. This step was required because machine learning algorithms require data in a structured form for effective analysis. By converting unstructured test responses into a tabular format, we ensure that each response has a consistent structure, with rows representing individual data points and columns representing features or attributes. After this pre-processing, the data were transformed into a tabular format with 18 columns, where each column represents a respective question. The responses are scored on a 5-point Likert scale (1—strongly disagree; 5—strongly agree). As supervised machine learning algorithms are employed, labeled data are required, and each survey in the dataset is tagged with a credit risk label. A binary label is used, where 0 represents low credit risk and 1 represents high credit risk.

After structuring the data, we proceeded to generate feature vectors. Feature vectors are essentially numerical representations of data points, making them digestible for machine learning models. In this process, each data point (in this context, a test response) is represented as a vector of numbers, where each number corresponds to a feature of the data. Here, each question in the survey corresponds to a specific attribute, which was numerically encoded using the target encoding method. After assigning questions to their respective attributes, feature vectors were created, and the corresponding numerical value of each answer was assigned to its corresponding attribute. For attributes with multiple related questions, the average of the answers is used to represent the value of that attribute.

After that, in order to find the impact of the attributes, a test should be applied, and a *t*-test has been selected for this study. The *t*-test is a statistical test method used to determine

if there is a statistically significant difference between the means of two groups. If the *p*-value is smaller than the predetermined level of significance (usually accepted as 0.05), then the null hypothesis (no difference between groups) is rejected, and a significant difference between the two groups is assumed. The *t*-test method was performed using the SciPy library in Python.

### 3.2. Target Encoding Module

Target encoding is an effective encoding technique for categorical variables and is often used in machine learning systems for processing tabular datasets with mixed numeric and categorical variables [50]. We argue that target encoding can help the model understand the relationship between the categorical variable and the target variable better than traditional feature encoding methods such as one-hot encoding. By encoding categories based on the average of the target variable, this approach might incorporate useful signals that improve the model's predictions. To this end, for this study, we chose the target encoding for the purposes of this study. We leave out the use of feature encoding methods such as one-hot encoding for future studies. In this study, each question in the survey corresponds to a specific attribute, which was numerically encoded using the target encoding method. After assigning questions to their respective attributes, a feature vector was created, and the corresponding numerical value of each answer was assigned to its corresponding attribute. For attributes with multiple related questions, the average of the answers is used to represent the value of that attribute. This process was applied to each survey in the dataset, resulting in a collection of feature vectors that constitute the dataset.

Target encoding is a key step in preparing the survey dataset for machine learning algorithms. By representing categorical attributes as numerical features, we can train models to predict credit risk based on the responses to the survey questions.

### 3.3. Model Construction Module

The model construction module is a crucial part of the proposed software architecture for predicting credit risk using machine learning algorithms. This module takes an input of labeled training data and produces a predictive model as an output.

There are five algorithms used in this module. The algorithms are decision tree, random forest, logistic regression, SVM, and KNN. Decision tree is another popular algorithm that builds a decision tree from the training data, and each node in the tree represents a decision rule based on the input features. Random forest is an ensemble learning method that constructs multiple decision trees during training and outputs the mode of the classes. SVM algorithm constructs a hyperplane that maximally separates the input space into different classes. Logistic regression is a regression-based algorithm that uses a logistic function to model the probability of a binary outcome. Finally, the KNN algorithm uses the distance between the input data and other training data to identify the nearest k neighbors and selects the most common class as the predicted label. We will train each algorithm on the labeled training data and evaluate their performance using metrics such as accuracy, precision, and recall. By comparing their performance, we can choose the most suitable algorithm for our credit risk assessment model.

### 3.4. Prediction Module

Once the model training is completed, predictions can be made using the trained model. A customer is presented with a questionnaire consisting of 18 questions. Based on the answers given to the questionnaire, a feature vector is created and used to make predictions, calculating the customer's credit risk.

## 4. Prototype and Evaluation

In this section, which technologies and libraries are used will be explained. As the programming language Python3 [51] was used for its readability and the availability of various libraries. The "csv" library was used to read the dataset. The scikit-learn library [52]

was used to train and test the dataset using algorithms such as decision tree and logistic regression, as mentioned above. Additionally, this library provides functions for splitting the dataset into training and testing sets.

The designed interface includes a menu where the model, dataset, and training/test ratios can be selected. The training is performed based on the selections made in this menu, and after the training, the window displays the results of accuracy, precision, recall, and f1 score. Furthermore, there is another page for customers, and this page contains survey questions. Based on the customers' responses, their credit risk is estimated. The PyQt5 library [53] was used to design this interface.

### 4.1. Dataset

In the data preparation phase, psychometric questions designed to measure an individual's creditworthiness were obtained from Aktif Bank Company. A dataset was then prepared, consisting of the psychometric questionnaire results of known customers with credit risk.

The dataset contains 18 questions, and each question is mapped to a corresponding feature. Figure 2 provides a detailed correlation matrix among various psychometric data features used in the analysis of credit risk prediction. Each cell in the matrix represents the correlation coefficient between a pair of features, with color intensities indicating the strength of the correlation—darker or brighter colors typically represent stronger correlations. Positive values denote a direct relationship, where an increase in one feature corresponds to an increase in the other, while negative values indicate an inverse relationship. This comprehensive visualization allows for immediate identification of features that are closely interrelated, as well as those that maintain independence, across the dataset.

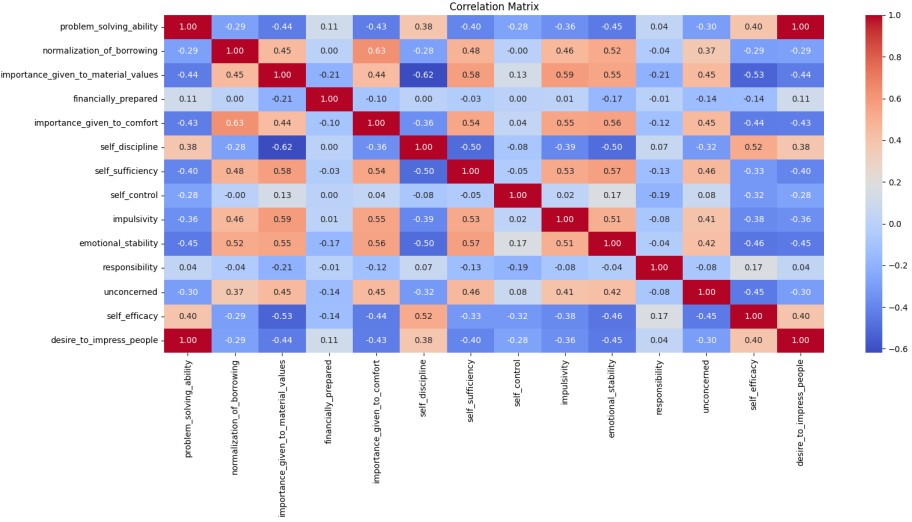

**Figure 2.** Correlation between the features.

Figure 3 showcases the correlation coefficients between various psychometric data features and the target class labels indicative of credit risk (e.g., "high risk" vs. "low risk"). Each feature is assigned a correlation value, ranging from −1 to +1, representing the strength and direction of the relationship with the target variable. Features with values closer to +1 or −1 have a strong positive or negative correlation with credit risk, respectively, while those near 0 show little to no correlation. The graph is visually designed to allow quick discernment of the most influential features, highlighting them in larger size.

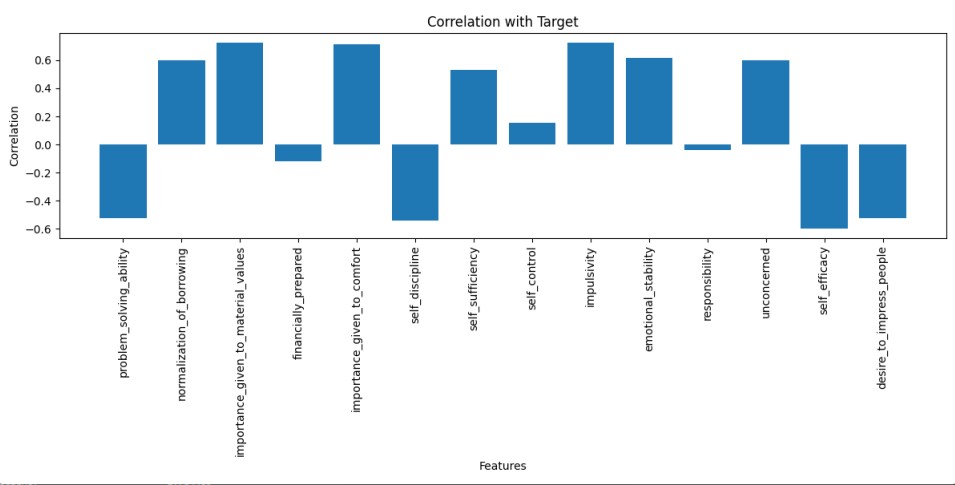

**Figure 3.** Correlation with target values.

Figure 4 presents the information gain values associated with each psychometric data feature used in predicting credit risk. Note that the feature importance values are derived from information gain metrics calculated through decision-tree-based models. While we employed decision-tree-based methodologies, we observed consistency in the ranking of feature importance across different individual decision-tree0based models. This consistency reaffirms the robustness of our selected features and the reliability of our methodology. Figure 4 illustrates a descending order of features, starting with the most impactful, based on the amount of information gain each contributes. Information gain is crucial as it quantifies how much each feature improves our ability to make accurate predictions, essentially indicating the reduction in entropy or uncertainty. In the context of credit risk, it highlights which aspects of a borrower's psychometric profile most significantly influence their likelihood of default or creditworthiness.

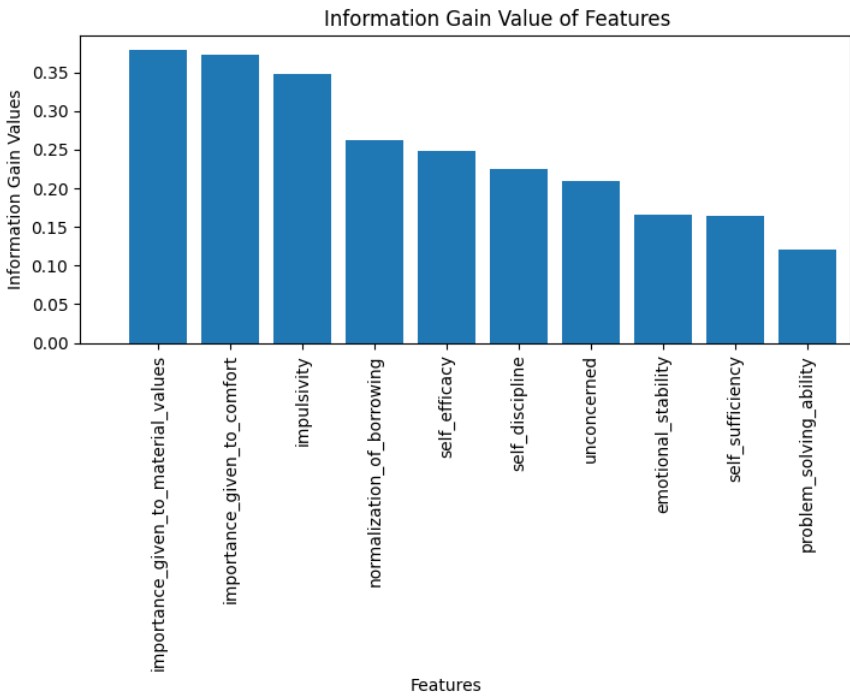

**Figure 4.** Information gain value of the top 10 features.

The features used in this study include attributes related to creditworthiness such as self-discipline, impulsivity, importance given to comfort, responsibility, and normalization

of borrowing. An attribute vector is created by calculating the average of the answers given to the questions representing each feature.

The target variable for this dataset represents the customer's credit risk, which ranges from 0 to 1. A value of 0 indicates low credit risk, while a value of 1 indicates high credit risk. Practitioners of data mining and machine learning have long observed that the imbalance of classes in a dataset negatively impacts the quality of classifiers trained on that data [54]. To mitigate this concern, we applied a stratified sampling technique during the data collection process. The ratio of positive (high credit risk, labeled as 1) to negative (low credit risk, labeled as 0) labels was maintained at approximately 45:55. This ensures that both classes are adequately represented in the dataset, allowing our models to better generalize and make more accurate predictions.

Table 1 provides a view of the descriptive statistics for the dependent feature, the credit risk score, across the dataset used in this study. It lists the average (mean), variance, and standard deviation, key metrics that together offer a snapshot of the distribution characteristics of the credit risk scores among the participants. The mean value indicates the average credit risk score, offering a central point around which the dataset is distributed. The variance and standard deviation provide insights into the dispersion of the scores, indicating the variability and consistency in the credit risk levels among the individuals assessed.

**Table 1.** Average, variance, and standard deviation values of dependent feature.

| Dependent Feature Name | Avg | Var | Std |
|---|---|---|---|
| Credit Risk Score | 0.4574 | 0.2508 | 0.5008 |

Table 2 delineates the methodology used for categorizing credit risk into discrete classes based on quantitative scores. It explicitly states the threshold values applied to the continuous credit risk scores, transforming them into binary outcomes: "high risk" and "low risk". According to the table, a pivotal point of 0.5 has been established as the benchmark—scores exceeding this value are classified as "high risk", indicating a less favorable credit assessment, while those falling below this threshold are deemed "low risk", suggesting a more favorable creditworthiness. This categorical approach simplifies the subsequent analysis and decision-making processes, providing actionable classifications based on quantifiable data.

**Table 2.** Threshold values of dependent feature.

| Dependent Feature Thresholds | Class Label |
|---|---|
| Credit Risk Score > 0.5 | High Risk |
| Credit Risk Score < 0.5 | Low Risk |

Table 3 presents a detailed breakdown of the dataset in terms of the number of instances classified under each category of the dependent feature, specifically the "high risk" and "low risk" credit scores. The table quantifies the exact number of occurrences or individuals falling into each group, thereby providing a depiction of the dataset's distribution.

**Table 3.** Number of dependent feature classes.

| Dependent Feature Classes | # of Instances |
|---|---|
| High Risk | 43 |
| Low Risk | 51 |

Table 4 compiles key descriptive statistics—namely, the average, variance, and standard deviation—for each independent feature in the dataset. These statistics offer foundational insights into the dataset's nature before any further modeling. The mean provides a central value for each feature, highlighting the average outcome that can be expected. Simultaneously, the variance and standard deviation deliver information about the spread of the data, indicating how much variability exists and how far data points tend to deviate from the mean, respectively.

**Table 4.** Average, variance, and standard deviation values of independent features.

|  | Independent Feature Name | Avg | Var | Std |  | Independent Feature Name | Avg | Var | Std |
|---|---|---|---|---|---|---|---|---|---|
| 1 | Problem Solving Ability | 3.2872 | 1.3467 | 1.1604 | 8 | Self Control | 2.8936 | 1.8165 | 1.3477 |
| 2 | Normalization of Borrowing | 3.0638 | 1.2915 | 1.1364 | 9 | Impulsivity | 2.9574 | 1.3960 | 1.1815 |
| 3 | Importance Given to Material Values | 2.8670 | 1.1514 | 1.0730 | 10 | Emotional Stability | 2.9042 | 1.5713 | 1.2535 |
| 4 | Financially Prepared | 3.3617 | 1.8677 | 1.3666 | 11 | Responsibility | 3.0106 | 1.9676 | 1.4027 |
| 5 | Importance Given to Comfort | 3.1542 | 1.4410 | 1.2004 | 12 | Unconcerned | 2.7978 | 1.2167 | 1.1030 |
| 6 | Self-Discipline | 3.2553 | 1.2459 | 1.1162 | 13 | Self-Efficacy | 3.1702 | 1.4330 | 1.1971 |
| 7 | Self Sufficiency | 3.0212 | 1.7844 | 1.3358 | 14 | Desire to Impress People | 3.2872 | 1.3467 | 1.1604 |

*4.2. Test Design*

In this study, the dataset has been split into training and testing datasets to evaluate the performance of the created model in an 80:20 ratio. At this point, attention was paid to the distribution of data for each label. The training is performed using several different algorithms, such as random forest, decision tree, and KNN, which are provided by the Python scikit-learn library. When adjusting the supervised machine learning model parameters, we conducted multiple experiments with different parameters for each algorithm. We observed that the results did not vary significantly with different parameters; generally, they yielded high-accuracy results. We believe that the similarity in performances is due to the nature of the dataset used. We argue that this might stem from the similarity in responses given by individuals to psychometric questions. The accuracy of the models is tested using metrics such as accuracy, precision, recall, and f1 score, which are calculated using algorithms provided by the scikit-learn library. For converting the probabilities into class labels (e.g., high risk or low risk in binary classification), we use the default threshold value, which is 0.5. If the predicted probability is greater than or equal to 0.5, the instance is classified as high risk; otherwise, it is classified as low risk. We calculated the accuracy, precision, recall, and f1 score metrics based on the default threshold value.

*4.3. Results*

Various metrics, such as accuracy, precision, recall, and f1 score, have been used to evaluate the performance of trained models.

The evaluation results shown in Table 5. Table 5 encapsulates the prediction performance results of the credit risk assessment models implemented in the study. It outlines key performance metrics, accuracy, precision, recall, FP rate, F1-score, and the area under the receiver operating characteristic curve (ROC). Each metric is important in evaluating the models' performance, providing insights into their reliability in correctly classifying instances into "high risk" and "low risk".

**Table 5.** Prediction performance results.

|  | Accuracy | Precision | Recall (TP Rate) | FP Rate | F1 Score | ROC Area |
|---|---|---|---|---|---|---|
| Logistic Regression | 0.9615 | 0.9736 | 0.9285 | 0.0666 | 0.9531 | 0.9309 |
| Random Forest | 0.9615 | 0.9736 | 0.8571 | 0.0 | 0.9531 | 0.9285 |
| Decision Tree | 0.9230 | 0.9500 | 0.9285 | 0.0666 | 0.9022 | 0.9309 |
| KNN | 0.9615 | 0.9736 | 0.8571 | 0.0 | 0.9531 | 0.9333 |
| SVM | 0.9615 | 0.9736 | 1.0 | 0.1333 | 0.9531 | 0.9333 |

Table 6 enumerates the results of the *t*-test analysis conducted on several key psychometric variables: self sufficiency, self-discipline, impulsivity, and self-control. Each of these attributes represents a dimension of behavior that could significantly influence an individual's financial decisions and credit risk.

**Table 6.** Results of the *t*-test.

| Financially Prepared | Self Sufficient | Self-Discipline | Impulsivity | Self Control |
|---|---|---|---|---|
| $5.8600 \times 10^{-8}$ | $6.2471 \times 10^{-33}$ | $6.2875 \times 10^{-26}$ | $1.8422 \times 10^{-33}$ | 0.00738 |

Table 7 provides a comprehensive evaluation of the prediction performance of the machine learning models used for assessing credit risk scores. This assessment is uniquely characterized by the prediction of continuous credit risk probabilities, a different approach compared to binary classifications. The table details critical metrics—including mean absolute error (MAE), root mean squared error (RMSE), and Matthew's correlation coefficient (MCC)—each reflecting different aspect of the model's accuracy and predictive power. The MAE quantifies the average magnitude of errors in the predictions, without considering their direction, offering a clear measure of accuracy on the continuous scale. The RMSE calculates the difference by squaring the errors before averaging them, thus placing more weight on large errors and indicating the presence of extreme variances in predictions. The MCC provides a balanced measure of the model's true positive and negative rates, indicating its ability to distinguish between different levels of risk accurately, even on a continuous scale. This table helps us to understand the model's effectiveness and reliability in predicting nuanced probabilities of credit risk, rather than just discrete categories.

**Table 7.** Prediction performance results—2.

| | Mean Absolute Error | Root Mean Squared Error | Matthew's Correlation Coefficient |
|---|---|---|---|
| Logistic Regression | 0.0522 | 0.1234 | 0.9080 |
| Random Forest | 0.1155 | 0.2390 | 0.8003 |
| Decision Tree | 0.0933 | 0.2115 | 0.8406 |
| KNN | 0.0422 | 0.0918 | 0.9264 |
| SVM | 0.0522 | 0.1234 | 0.9080 |

Accuracy is a common metric used to assess the overall correctness of a classification model's predictions. It measures the proportion of correctly classified instances (both true positives and true negatives) out of all instances. The random forest, KNN, SVM, and logistic regression algorithms achieved the same level of accuracy at 96.15%. The decision tree algorithm achieved a slightly lower accuracy of 92.30%. Therefore, based on these results, we conclude that the evaluated machine learning algorithms are suitable for predicting a customer's credit risk via psychometric tests with high accuracy. These values reflect the models' overall classification capabilities and are indicative of their quality in correctly classifying instances in our binary classification task.

Precision is a metric that measures the proportion of true positive predictions (correctly identified positive cases) out of all instances predicted as positive. It provides insight into the model's ability to make accurate positive predictions without producing many false positives. In summary, in this study, the precision values suggest that all of the supervised learning models perform well in terms of making accurate positive predictions while minimizing false positives, with the logistic regression, random forest, KNN, and SVM models achieving the highest precision scores.

Recall, also known as sensitivity or true positive rate (TP Rate), measures the proportion of true positive predictions (correctly identified positive cases) out of all actual positive cases. It provides insight into the model's ability to correctly identify positive instances. In summary, the recall values suggest that all of the models, used in this study, perform well in terms of capturing and correctly identifying positive instances, with the SVM model

achieving the highest recall score, while the logistic regression and decision tree models achieved slightly lower but still high scores.

The F1 Score is a metric that combines precision and recall into a single value, providing a balanced measure of a model's overall performance in binary classification tasks. In summary, in this study, the F1 Score values suggest that all of the supervised learning models provide a balanced and effective performance in terms of correctly identifying positive cases while minimizing both false positives and false negatives.

The false positive rate (FP Rate), measures the proportion of false positive predictions (incorrectly identified positive cases) out of all actual negative cases. It reflects a model's tendency to make false positive errors. Based on the results, the FP Rate values indicate that the random forest and KNN models are particularly effective at avoiding false positive errors, with the SVM model having a slightly higher rate of false positives.

The area under the receiver operating characteristic curve (ROC Area) is a metric used to evaluate the overall performance of a binary classification model. It measures the model's ability to distinguish between positive and negative cases across different probability thresholds. In this study, The ROC area metric values suggest that all of the supervised learning models perform well in terms of their ability to discriminate between positive and negative cases, with the KNN and SVM models achieving the highest scores.

The notable high performance of these algorithms underscores the effectiveness of the selected features in capturing credit risk accurately. This suggests that the features used in this study successfully capture the relevant information needed to assess credit risk. Based on these results, the developed system holds promise as a supportive tool for credit risk prediction.

Based on the results shown in Table 6, it can be observed that the *p*-values are significantly smaller than 0.05, indicating that these features have an effect on the dependent variable.

To ensure the selection of relevant attributes, it is important to test whether these attributes have an impact on the dependent variable (high or low credit risk), and having attributes that do not reflect the customer's credit risk would complicate the survey. For this purpose, the statistical analysis method "*t*-test" has been applied. If the *p*-value is smaller than the predetermined level of significance (usually accepted as 0.05), then a significant difference between the two groups is accepted. An example of the results for self-discipline has a *p*-value of $6.2875 \times 10^{-26}$. Additionally, the *p*-value for the impulsivity attribute was found to be $1.8422 \times 10^{-33}$, and the *p*-value for the self-control attribute was 0.00738. Thus, the appropriateness of the attributes has been confirmed.

Note that the results of our model are probabilistic estimates. To turn them into binary classifications, the error in these continuous forecasts had to be measured. To this end, in our experimental study, we went beyond the usual level of classification accuracy and looked into how wrong the predictions were. With this analysis, we could see more details about how well the model worked with this method, which helped us figure out not only if an estimate was wrong, but also by how much.

In this study, we investigate the effectiveness of our approach from the perspective of mean absolute error (MAE), root mean squared error (RMSE), and Matthew's correlation coefficient (MCC) values for each model. The results are listed in Table 7.

Based on the results, it appears for the MAE metric, all of our machine learning models are performing quite well. MAE measures the average absolute difference between the predicted values and the actual values. Hence, on average, these models' predictions are negligible units away from the actual values.

RMSE measures the average magnitude of the errors between predicted values and actual values, with a focus on larger errors. Based the results, an RMSE indicates that, on average, the models' predictions deviate negligible units from the actual values. This also indicates that these models perform well on the dataset.

MCC is a metric used primarily for binary classification tasks and assesses the overall quality of binary classification predictions. It takes into account true positives, true negatives, false positives, and false negatives. Higher MCC values indicate better model

performance, specifically in terms of correctly classifying data points into the correct binary classes. Based on the results, the MCC values are relatively high across all models, suggesting that these models are performing well in terms of classification accuracy.

## 5. Conclusions and Future Work

In this study, we examined the effectiveness of using psychometric tests in the credit lending process to measure customers' creditworthiness. We found that utilizing psychometric testing data is a promising approach for credit lending decisions. Furthermore, it is revealed that the use of psychometric tests can help with financial inclusion by allowing customers without a credit history or sufficient data to access financial opportunities.

Supervised machine learning algorithms, such as logistic regression, random forest, and decision tree, were used to predict credit risk. The results show that the psychometric data that we use can be used to predict the likelihood of loan default with reasonable accuracy.

Within the scope of this research, the results obtained from our experimental study are dependent on the dataset we used. Additionally, the results we obtained are also contingent upon the machine learning algorithms we employed. It should be noted that using a different dataset or employing alternative algorithms may lead to varying outcomes.

Future work can explore the effectiveness of using other machine learning algorithms and models to predict credit risk based on psychometric data. Additionally, the impact of using psychometric tests on customer satisfaction and the time added to the credit lending process can be further examined. Further research can also investigate the impact of other variables, such as demographic information, on credit risk prediction.

**Author Contributions:** Conceptualization, E.D., E.Y. and M.S.A.; methodology, E.D., E.Y. and M.S.A.; data curation and software, E.D.; writing, reviewing and editing, E.D., E.Y. and M.S.A.; supervision, M.S.A. All authors have read and agreed to the published version of the manuscript.

**Funding:** This research received no external funding.

**Data Availability Statement:** The authors agree to share datasets upon requests from readers.

**Acknowledgments:** The authors acknowledge Aktifbank for their support, encompassing essential research facilities, datatests, advanced computing platforms, and an enriching working environment, all of which significantly contributed to the successful completion of this study.

**Conflicts of Interest:** The authors declare no conflict of interest.

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
