# Peer review of "Credit Risk Prediction Based on Psychometric Data"

_computers, doi:10.3390/computers12120248_

Round 1

Reviewer 1 Report

Comments and Suggestions for Authors

14 'likelihood of loan' You probably mean the likelihood of a loan being repaid.

Figure 1 'Target Encoding' is underlined in red.

189 users behaviors -> users' behaviors

275 There is no space after the full stop.

Comments on the Quality of English Language

The English is good.

Author Response

We would like to thank the reviewer for helping us to improve the quality of the content. We updated the manuscript according to the comments and suggestions. 

Answers to the Comments and Suggestions by Reviewers

Comment-1: 14 'likelihood of loan' You probably mean the likelihood of a loan being repaid.
Answer-1: Yes, that is what we meant. Thanks! We updated the statement to make it more clear like the
reviewer suggested.
Comment-2: Figure 1 'Target Encoding' is underlined in red.
Answer-2: Thanks! We fixed it!
Comment-3: 189 users behaviors -> users' behaviors
Answer-3: Thanks. We fixed it!
Comment-4: 275 There is no space after the full stop.
Answer-4: Thanks. We fixed it!

Reviewer 2 Report

Comments and Suggestions for Authors

Full Title: Credit Risk Prediction Based on Behavioral Data Analysis

Objective: The present study aimed to propose a methodology to perform a credit risk prediction based on behavioral data analysis.

There are some issues that need to be addressed:

1.     First of all, author(s) should check English. I suggest the authors to proofreading their text. There are many minor mistakes.

2.     Please list and explain the features (dependent and independent variables) of the prediction model. Provide description of the data set in the Method section.

3.     Providing a predictive model in a figure (TIFF images with minimum 300 dpi) showing the relationships between study variables would be better.

4.     Provide descriptive statistics (f, %) about the classes of the dependent variable (customer’s credit risk).

5.     Likewise, provide descriptive statistics (means, SD) about the features or attributes (independent variables) of the model.

6.     Provide classifiers’ performance indicators (CCI, TP Rate, FP Rate, Precision, Recall, F-Measure, MCC, ROC Area) in a table.

7.     Provide Mean Absolute Error (MAE), Root Mean Squared Error (RMSE), and Matthew’s Correlation Coefficient (MCC) values for each model. RMSE is a quadratic metric that measures the size of the error that is often used to find the distance between the predicted values and the observed values of the estimator. Therefore, it is important to report RMSE to properly assess the performance of the model.

8.     Please conduct a sensitivity analysis to identify the most important input factors.

Comments on the Quality of English Language

1.   Author(s) should check English. I suggest the authors to proofreading their text. There are many minor mistakes.

Author Response

We want to thank the reviewer for helping us improve the paper's quality. We addressed the reviewer's comments and updated the manuscript according to the comments/suggestions made by the reviewer. 

Reviewer 3 Report

Comments and Suggestions for Authors

Credit Risk Prediction Based on Behavioral Data Analysis

To handle the loan applicants without credit profiles, this manuscript has evaluated several classical machine learning models based on psychometric features for the purpose of credit risk prediction. This is an interesting topic, but a few questions need to be clarified in the manuscript before acceptance. 

Minor comments:

  1. Title: ‘behavioral data analysis’ is probably not appropriate here because the data in this work is not related to user behavior and there is no analysis work. I would suggest directly mentioning ‘psychometric data’ rather than ‘behavioral data analysis’

  2. Line 77: duplicate ‘responsibility’

  3. Line 232: I think ‘Random Classifier’ has not been used in this work, please fix the corresponding paragraph

  4. Line 321-322: the work does not show improvement by psychometric data, otherwise you need ablation tests, such as comparing model performance with and without psychometric data.

  5. Line 251-262: packages need references 

General comments:

  1. Preprocessing and data overview: any preprocessing has been performed before training the models? Could you please show some basic statistics of the feature values? Mean, variance, correlation between features, correlation with target values

  2. Target Encoding: why using target encoding rather than directly using feature encoding such one-hot encoding?

  3. Dataset: please provide more explanation about ‘paying attention to the proportions of target variables’. Is it stratified sampling? What is the ratio of positive and negative labels? 

  4. Model tuning: how were models tuned? 4 of the models performed exactly the same, any reasons? 

  5. Metrics: what is the probability threshold for accuracy, precision and recall?

  6. Explainability: for logistic regression, RF and DT models, could you show the feature importance?

Author Response

(The authors gave the same response as above.)

Round 2

Reviewer 2 Report

Comments and Suggestions for Authors

Thank you to the authors for their prompt response. They have thoroughly addressed all of my concerns.

Author Response

Reviewer's Comment: Thank you to the authors for their prompt response. They have thoroughly addressed all of my concerns.

Author's Response: We thank the reviewer for all the help and guidance.  We greatly appreciate it. 

Reviewer 3 Report

Comments and Suggestions for Authors

1. Some typos such as 'future classes'

3. Many tables and figures do not have any description, please add your insight in paragraph.

4. feature importance: which model does this come from? any difference there between different feature importance?

5. The revised line 428-438 is confused to me, why use MAE and and RMSE to evaluate classification models? 

Comments on the Quality of English Language

English language is fine.

Author Response

We thank the reviewer for helping us improve the paper's quality. We addressed the reviewer's comments and updated the manuscript according to the comments and suggestions made by the reviewer. We added an Author's Notes File and explained what we did for each suggestion.
